

# Generating the electro-weak scale by vector-like quark condensation

Sophie Klett[*], Manfred Lindner[†] and Andreas Trautner[‡]

Max-Planck-Institut für Kernphysik, Saupfercheckweg 1, 69117 Heidelberg, Germany

[*] sophie.klett@mpi-hd.mpg.de , [†] lindner@mpi-hd.mpg.de ,
[‡] trautner@mpi-hd.mpg.de

## Abstract

We show that vector-like quarks in the fundamental or higher-dimensional representations of QCD can generate the electro-weak scale in a phenomenologically viable way by chiral symmetry breaking condensates. The thereby generated scales are determined by numerically solving the Dyson-Schwinger equation and these scales are sizable, because they grow with the hard vector-like mass. Communicating such a scale to the Standard Model via a conformally invariant scalar sector can dynamically generate the electro-weak scale without a naturalness problem, because all non-dynamical mass scales are protected by chiral symmetry. We present a minimal setup which requires only a new neutral scalar with mass not too far above the electro-weak scale, as well as vector-like quarks at the (multi-)TeV scale. Both are consistent with current bounds and are attractive for future experimental searches at the LHC and future colliders. Depending on the hypercharge of the vector-like quarks, hadrons made of them are color-neutral bound states which would be interesting Dark Matter candidates.



# 1 Introduction

For many years supersymmetry was expected to be the solution to the hierarchy problem. Meanwhile supersymmetric explanations are pushed to specific corners of parameter space suggesting that other mechanisms may be at work. An interesting direction is that the hierarchy problem might be related to scale invariance which is only broken at the quantum level. The dynamical generation of the electro-weak (EW) scale can then either be realized via a Coleman-Weinberg mechanism [1–5] or by dimensional transmutation in a strongly interacting sector. Concerning the latter approach, many models rely on additional hidden gauge groups [6–9] while others rely only on standard quantum chromodynamics (QCD) (see e.g. [10]). This last scenario is particularly tempting as it does not require a further gauge extension of the SM. In fact, the idea of generating the EW scale from non-perturbative QCD effects exists for a while. Already in the 1980s, the authors of [11–14] suggested to break the EW symmetry by the condensation of chiral fermions in high color representations. They conjectured that exotic fermion condensates generate larger scales than the usual triplet representation since the criticality condition

$$C_2(\mathbf{R})\,\alpha_s(\Lambda) \gtrsim \mathcal{O}(1)\,,$$

is fulfilled for smaller values of the strong coupling $\alpha_s$ owing to a larger Casimir Constant $C_2(\mathbf{R})$ of a higher representation $\mathbf{R}$. Albeit an interesting and natural mechanism, the original idea is ruled out from EW precision observables [15, 16].

We suggest in this paper a modified scenario with a vector-like (VL) fermion being a singlet under the EW gauge group but charged under $SU(3)_C$. The chiral symmetry breaking condensate $\langle\overline{\psi}\psi\rangle \neq 0$ can then induce a vacuum expectation value (VEV) for the SM Higgs $\phi$ by a singlet scalar mediator $S$. Thus, EW symmetry breaking (EWSB) is triggered indirectly via the scalar portal. This mechanism enables the dynamical generation of the EW scale, despite starting from a classically scale invariant scalar sector. A realistic model requires that the explicit VL fermion mass is $\gtrsim \mathcal{O}(1\,\mathrm{TeV})$ to escape current direct detection limits. Although this introduces an explicit scale to the model, it can be thought of as being generated in an enlarged scale-invariant setting. Note that such a VL mass is technically natural as chiral symmetry protects the fermionic mass term [17]. The main part of this work is devoted to the unusual dynamics of the problem, i.e. to determine the condensate of a massive VL fermion. Instead of relying on the approximate gap equation, we study the condensate and dynamical chiral symmetry breaking (DCSB) in the framework of Dyson-Schwinger equations (DSEs).

The paper is organized as follows:. In Section 2 we recapitulate the fermion DSE and the truncation scheme that is used to solve the equation. Furthermore, we introduce the formalism to extract the quark condensate beyond chiral limit and apply this to more general representations of the color gauge group in Section 3. In Section 4 we outline how the quark condensate can induce EWSB. Finally, we summarize our results in Section 5.

# 2 Fermion Condensate Beyond the Chiral Limit

To study the properties of the quark condensate beyond the chiral limit, we solve the DSE for the quark propagator [18–20]. In its renormalized form, it is given by

$$S^{-1}(p) = Z_2\left(\not{p} - Z_m\, m_\mu\right) - \mathrm{i}\, C_2(\mathbf{R})\, Z_{1\mathrm{F}}\, g^2 \int \frac{d^4 k}{(2\pi)^4}\, \gamma_\mu\, S(k)\, \Gamma_\nu(k,p)\, D^{\mu\nu}(p-k)\,, \qquad (1)$$

where the renormalization constants for quark wavefunction, quark-gluon vertex and mass are denoted by $Z_2$, $Z_{1\mathrm{F}}$ and $Z_m$, respectively. $m_\mu$ indicates the renormalized current quark mass at

the renormalization scale $\mu$ and $C_2(\mathbf{R})$ is the Casimir invariant for a quark in representation $\mathbf{R}$. The DSE for the quark propagator depends on both, the full gluon propagator $D^{\mu\nu}$ and the dressed quark-gluon vertex $\Gamma_\nu$. These, in turn, fulfill their own DSEs which together form a system of coupled differential equations. Since we do not want to solve these equations simultaneously, we decouple the system by setting $\Gamma_\mu(k,p) = \gamma_\mu$, which is also referred to as the "rainbow approximation" [19]. Furthermore, we substitute the factor

$$Z_{1F}\, g^2\, D^{\mu\nu}(p-k) \longrightarrow 4\pi\,\alpha_{\text{eff}}\left((p-k)^2\right) D_0^{\mu\nu}(p-k)\,, \tag{2}$$

where $D_0^{\mu\nu}(p) := p^{-2}\left(g^{\mu\nu} - p^\mu p^\nu p^{-2}\right)$ is the free gluon propagator in Landau gauge and $\alpha_{\text{eff}}(k^2)$ is an effective strong coupling evaluated at a scale $k^2$ [18]. Under these assumptions, all non-perturbative effects of the dressed gluon propagator are completely incorporated by a phenomenologically motivated effective running coupling. For our study, we follow [21, 22] and use an effective running coupling which is given by

$$\alpha_{\text{eff}}(k^2) = 2\pi\,\frac{D}{\omega^4}\,k^2 \exp\left(-\frac{k^2}{\omega^2}\right) + \frac{2\pi\gamma_m}{\ln\left[\tau + (1+\frac{k^2}{\Lambda_{\text{QCD}}^2})^2\right]}\left[1 - \exp\left(\frac{-k^2}{4m_\perp^2}\right)\right]\,, \tag{3}$$

with $m_\perp = 0.5\,\text{GeV}$, $\tau := e^2 - 1$, $\Lambda_{\text{QCD}} = 0.234\,\text{GeV}$, $\gamma_m = \frac{12}{33-2n_F}$, and $n_F = 6$. The parameters $\omega$ and $D$ are responsible for the low momentum behavior and can be chosen in such a way that there is enough integrated strength in the infrared (IR) for dynamical chiral symmetry breaking to happen.[1] For large momenta, $\alpha_{\text{eff}}$ reproduces the perturbative QCD running. Studies aiming to fit observables in the vector and pseudoscalar meson sector have shown that there is a good agreement with observations for $\omega \in [0.4, 0.6]\,\text{GeV}$, which is almost unaffected by parametric variations as long as $(\omega D) = (0.8\,\text{GeV})^3$ is kept constant [24]. In the following we will, therefore, employ the commonly used values $\omega = 0.5\,\text{GeV}$ and $D = 1.024\,\text{GeV}^2$.

The solution to Eq. (1) generally takes the form

$$S^{-1}(p) \equiv Z^{-1}(p^2)\left[\slashed{p} - M(p^2)\right]\,, \tag{4}$$

with $M(p^2)$ being the dynamical mass function and $Z(p^2)$ the quark wave function renormalization. For our numerical study, we choose a momentum subtraction renormalization scheme at $\mu^2 = \Lambda^2$, where $\Lambda$ is a momentum cutoff. This is a convenient choice of renormalization scale since all renormalization constants are approximately one and we can neglect them from now on. At the renormalization scale $\mu^2$, the boundary condition is given by $S^{-1}(p)\big|_{p^2=\mu^2} \simeq \slashed{p} - m_\mu$. Thus, at momenta $p^2 = \mu^2$ the fermion self-energy vanishes, i.e. $M(p^2 = \mu^2) = m_\mu$ and $Z(p^2 = \mu^2) = 1$. In the subsequent calculations we chose $\Lambda^2 = (10^6\,\text{GeV})^2$ which is well above the mass scale of a few TeV which we are interested in. The numerical calculation is carried out in discretized momentum space with $N = 500$ sample points and an infrared cutoff $p^2 = 10^{-4}\,\text{GeV}^2$, and we use Gauss-Legendre quadrature in order to solve the discrete momentum integrals [25]. The resulting wave function renormalization and dynamical mass functions are shown in Fig. 1 for a representative set of different current masses.

For quarks with zero current mass the chiral condensate is defined as the trace of the quark propagator in Dirac and color space [26, 27]

$$-\langle\overline{\psi}\psi\rangle_\mu = \lim_{x\to 0}\text{Tr}\left[S(x)_{m_\mu=0}\right] = \frac{d(\mathbf{R})}{4\pi^2}\int dk^2\,\frac{k^2 Z(k^2) M(k^2)}{k^2 + M^2(k^2)}\,. \tag{5}$$

Here, $S(x)$ is a position space solution to the DSE for $m_\mu = 0$ and $d(\mathbf{R})$ is the dimension of the quark representation under the color gauge group. In the chiral limit, the integral in

---

[1]Studies of weakly coupled theories like e.g. quantum electrodynamics [23] showed that there is a critical coupling below which there is no dynamical chiral symmetry breaking and, hence, no condensate being generated.

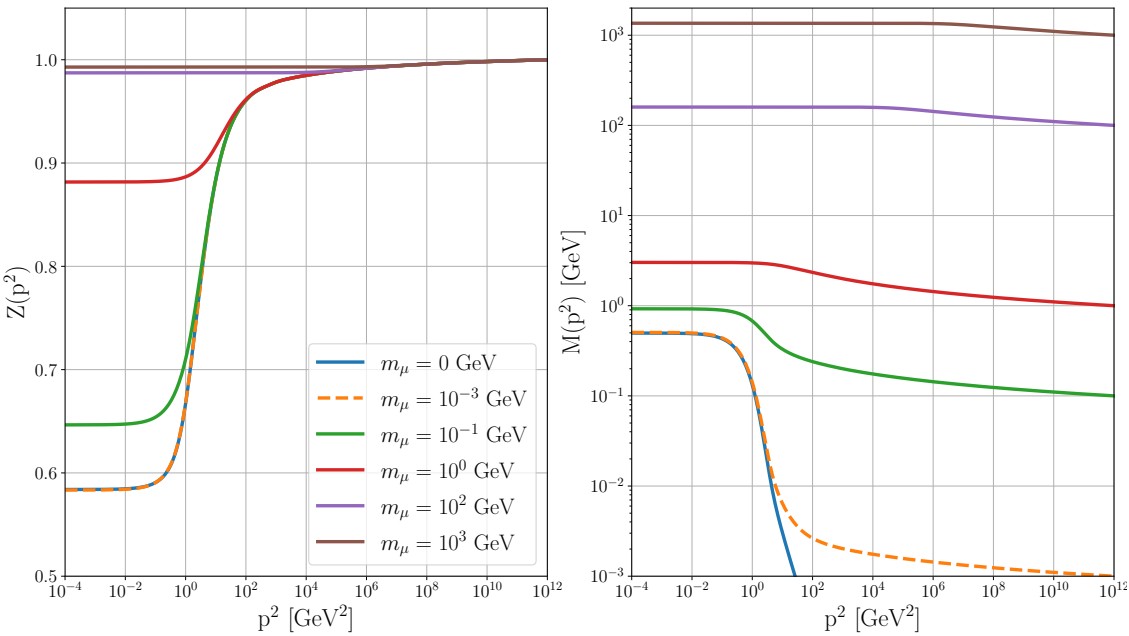

Figure 1: Wave function renormalization $Z(p^2)$ (left) and mass function $M(p^2)$ (right) for fermions in a triplet representation with different current masses $m_\mu$, defined at the scale $\mu = 10^6$ GeV. For comparison, the chiral limit $m_\mu = 0$ is shown as well (blue line).

Eq. (5) is well defined and finite due to the rapidly decreasing dynamical mass function for large momenta (see Fig. 1). To understand the short distance behavior, it is helpful to consider the operator product expansion (OPE) [28–30]. According to the OPE, the dynamical mass function in the limit $p \to \infty$ behaves like

$$M(p^2) \simeq \hat{m} \left[ \frac{1}{2} \ln\left( \frac{p^2}{\Lambda_{\text{QCD}}^2} \right) \right]^{-\gamma_m} - \frac{2\pi^2 \gamma_m}{d(\mathbf{R})} \frac{\langle \overline{\psi}\psi \rangle_{\text{inv}}}{p^2} \left[ \frac{1}{2} \ln\left( \frac{p^2}{\Lambda_{\text{QCD}}^2} \right) \right]^{\gamma_m - 1}, \quad (6)$$

where we have defined the renormalization group invariant quantities

$$\hat{m} := m_\mu \left[ \frac{1}{2} \ln\left( \frac{\mu^2}{\Lambda_{\text{QCD}}^2} \right) \right]^{\gamma_m}, \quad \text{and} \quad \langle \overline{\psi}\psi \rangle_{\text{inv}} := \langle \overline{\psi}\psi \rangle_\mu \left[ \frac{1}{2} \ln\left( \frac{\mu^2}{\Lambda_{\text{QCD}}^2} \right) \right]^{-\gamma_m}. \quad (7)$$

Whereas the coefficient of the operator $\langle \overline{\psi}\psi \rangle_{\text{inv}}$ decays as $p^{-2}$, the current mass is only subject to a logarithmic running. Hence, it is the large momentum behavior that provides a basic distinction between the explicit symmetry breaking mass and the dynamically generated condensate.

Applying Eq. (5) to our solution for $M(p^2)$ and $Z(p^2)$ we find $-\langle \overline{\psi}\psi \rangle_{\text{inv}} = (0.218 \, \text{GeV})^3$ for a chiral quark in the color triplet representation. For massive quarks, the definition (5) via the trace of the propagator cannot simply be applied since the integral contains divergences induced by the term linear in the current quark mass [31]. This fact can be illustrated by considering the effect of an explicit mass term $m$ that contributes to $M(p^2)$. Approximating $Z(p^2) = 1$, the condensate would become

$$-\langle \overline{\psi}\psi \rangle_\mu \sim \int^{\Lambda^2} dk^2 \frac{k^2 m}{k^2 + m^2} = m \Lambda^2 + m^3 \ln\left( \frac{m^2}{\Lambda^2 + m^2} \right), \quad (8)$$

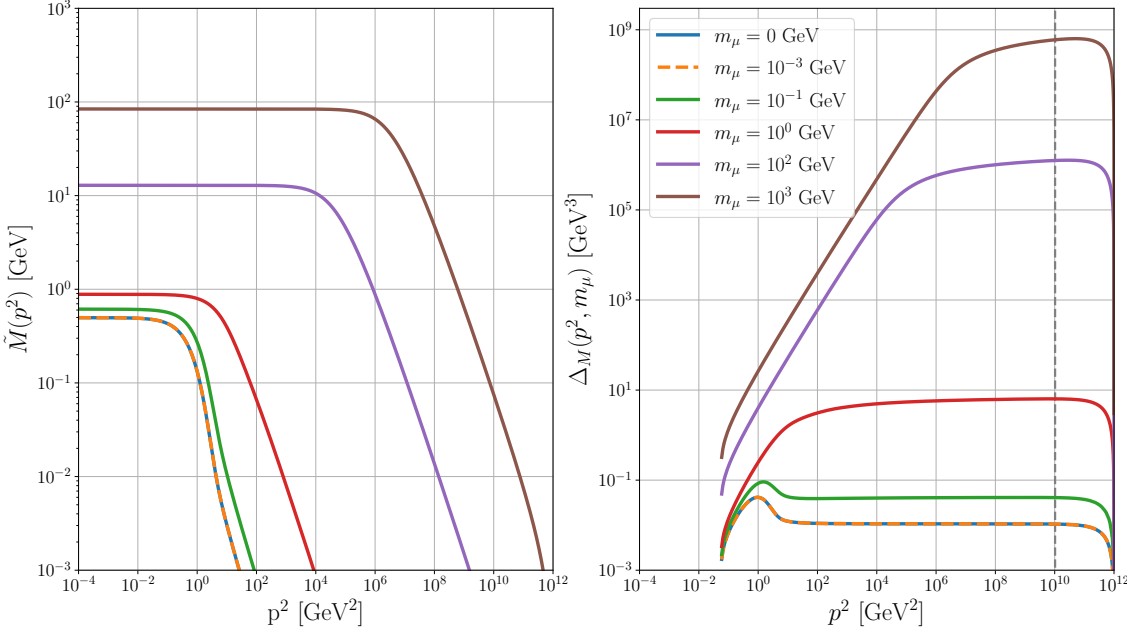

Figure 2: $\tilde{M}(p^2)$ (left) and $\Delta_M(p^2, m_\mu)$ (right) for fermions in a color triplet representation with different current masses. The dashed vertical line in the right plot indicates the momentum that we use to determine the value $C(m_\mu)$.

which clearly exposes the appearing quadratic and logarithmic divergences as a function of the momentum cutoff $\Lambda^2$. An apparently straightforward alternative to using Eq. (5) would be to employ the OPE for $M(p^2)$ and extract the condensate from the coefficient of the term proportional to $p^{-2}$. For instance, the authors of [21] attempted to fit Eq. (6) to their numerical solution of the DSE. However, this turns out to be a delicate task that cannot simply be performed for quarks which are heavier than the strange quark due to their dominating explicit mass terms. There are several other proposals for an unambiguous definition for the condensate of massive fermions, which try to cancel the divergent parts of the integral in Eq. (8) by subtracting either a wisely chosen fraction of the strange quark condensate [32–34] or the derivative of the condensate [35]. In our work we will follow the latter approach because we are interested in current quark masses of much higher scales.

Evidently, the term linear in $m_\mu$ can be removed from Eq. (6) by considering the redefined quantity

$$\tilde{M}(p^2) := \left(1 - m_\mu \frac{d}{dm_\mu}\right) M(p^2). \tag{9}$$

According to the OPE this should scale like

$$\tilde{M}(p^2) \overset{p\to\infty}{\simeq} -\frac{2\pi^2 \gamma_m}{d(\mathbf{R})} \frac{C(m_\mu)}{p^2} \left[\frac{1}{2} \ln\left(\frac{p^2}{\Lambda_{\text{QCD}}^2}\right)\right]^{\gamma_m - 1}. \tag{10}$$

Here we have defined the function

$$C(m_\mu) := \left(1 - m_\mu \frac{d}{dm_\mu}\right) \langle \overline{\psi}\psi \rangle_{\text{inv}}^{m_\mu}, \tag{11}$$

where the new dependency of the condensate $\langle \overline{\psi}\psi \rangle_{\text{inv}}^{m_\mu}$ on the current quark mass $m_\mu$ is indicated by a superscript. The left plot in Fig. 2 illustrates the behavior of $\tilde{M}(p^2)$ for different
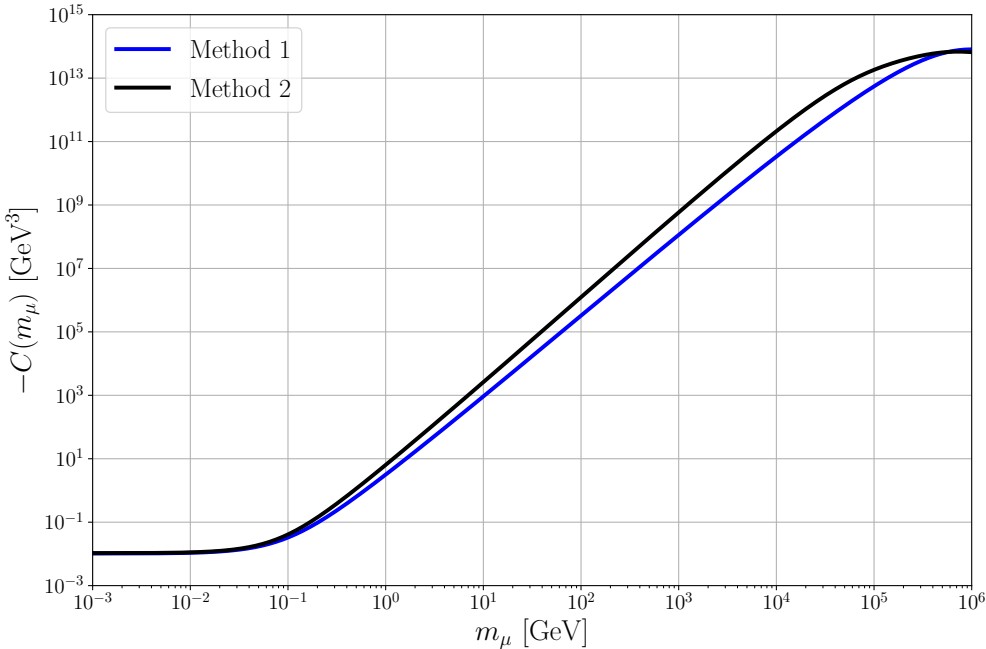

Figure 3: The order parameter of chiral symmetry breaking in the infrared, $C(m_\mu)$, as a function of the current quark mass, $m_\mu$, for fermions in the color triplet representation. The two different methods we use to extract this behavior (see text) are in reasonable agreement.

masses, which agrees well with the expected behavior from Eq. (10). Using these considerations, we apply two different methods to isolate the momentum independent quantity $C(m_\mu)$ from $\tilde{M}(p^2)$.

**Method 1:** As $\tilde{M}(p^2)$, by definition, does not depend linearly on $m_\mu$, none of the divergences shown in Eq. (8) emerge and we can safely evaluate the integral on the right side of Eq. (5) with $M(p^2)$ replaced by $\tilde{M}(p^2)$. This results in

$$\frac{d(\mathbf{R})}{4\pi^2} \int^{\Lambda^2} dk^2 \frac{k^2 Z(k^2) \tilde{M}(k^2)}{k^2 + \tilde{M}^2(k^2)} \overset{\text{Eq. (10)}}{\simeq} -C(m_\mu) \left[\frac{1}{2} \ln\left(\frac{\Lambda^2}{\Lambda_{\text{QCD}}^2}\right)\right]^{\gamma_m}, \qquad (12)$$

where we again used the OPE and divide by the factor $[1/2 \ln(\Lambda^2/\Lambda_{\text{QCD}}^2)]^{\gamma_m}$ to remove it from the right hand side in order to obtain $-C(m_\mu)$.

**Method 2:** Alternatively, we can infer $C(m_\mu)$ from the large momentum behavior of $\tilde{M}(p^2)$. From Eq. (10) it follows that the quantity

$$\Delta_M(p^2, m_\mu) := p^2 \tilde{M}(p^2) \frac{d(\mathbf{R})}{2\pi^2 \gamma_m} \left[\frac{1}{2} \ln\left(\frac{p^2}{\Lambda_{\text{QCD}}^2}\right)\right]^{-(\gamma_m - 1)} \xrightarrow{p \to \infty} -C(m_\mu), \qquad (13)$$

is a constant in momentum for $p \to \infty$. We can, therefore, derive $-C(m_\mu)$ from the plateau region of $\Delta_M(p^2, m_\mu)$ (see right plot in Fig. 2).

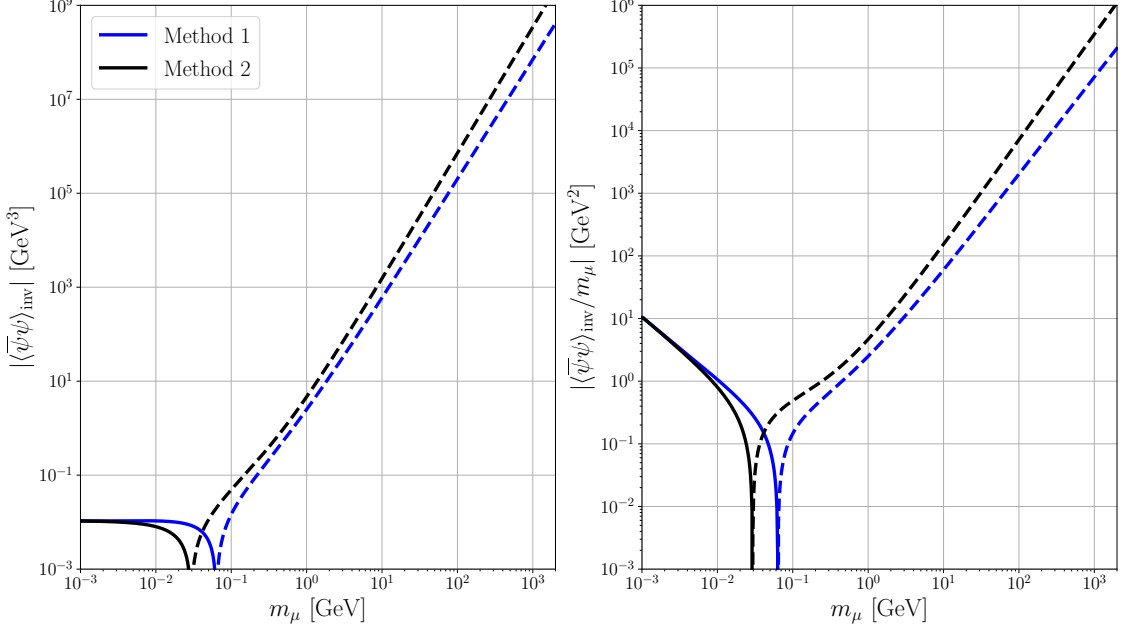

Figure 4: Absolute value of the condensate $|\langle \overline{\psi}\psi \rangle_{\text{inv}}|$ (left) and the ratio $|\langle \overline{\psi}\psi \rangle_{\text{inv}}/m_\mu|$ (right) as a function of the fermion current mass $m_\mu$ for fermions in the color triplet representation. The blue and black lines show two different methods to extract the condensate which are in reasonable agreement and show the theoretical error involved. The solid(dashed) line shows regions where $\langle \overline{\psi}\psi \rangle_{\text{inv}} < 0$ ($\langle \overline{\psi}\psi \rangle_{\text{inv}} > 0$).

The result of our evaluation for both methods are shown in Fig. 3. Evidently, both methods agree to a level that is more than sufficient for our purpose here. In the limit $m_\mu \to 0$ our result reproduces the chiral condensate obtained from Eq. (5), while $-C(m_\mu)$ increases for larger masses. A similar behavior was found only recently by [36]. We note that the kink for masses near $10^5$ GeV is a cutoff effect that will not affect our final result as we are mostly interested in current masses of a few TeV. In any case, the effect could easily be overcome by choosing a larger $\Lambda$.

Earlier studies of massive quark condensates typically stop with $C(m_\mu)$ as the final order parameter of chiral symmetry breaking [32,33,35]. However, for our model we are interested in the expectation value of the fermion two point function, which is precisely the expansion coefficient $\langle \overline{\psi}\psi \rangle_{\text{inv}}^{m_\mu}$ of the OPE. This is why we proceed by extracting $\langle \overline{\psi}\psi \rangle_{\text{inv}}^{m_\mu}$ from the obtained $C(m_\mu)$. From the definition, Eq. (11), we know that

$$-\frac{C(m_\mu)}{m_\mu^2} = \frac{d}{dm_\mu}\left[ \frac{\langle \overline{\psi}\psi \rangle_{\text{inv}}^{m_\mu}}{m_\mu} \right]. \tag{14}$$

Integrating over $m_\mu$ then gives

$$-\frac{\langle \overline{\psi}\psi \rangle_{\text{inv}}^{m_\mu}}{m_\mu} = \left[ -\frac{\langle \overline{\psi}\psi \rangle_{\text{inv}}^{m_\mu}}{m_\mu} \right]_{m_\mu = \epsilon} + \int_\epsilon^{m_\mu} \frac{C(m_\mu)}{m_\mu^2}\, dm_\mu, \tag{15}$$

where $\epsilon$ is a small current mass that we use as a reference point. We assume that the condensate of a light quark is essentially the same as that of a chiral quark. Therefore, for quarks in the triplet representation, we choose boundary conditions $\epsilon = 0.001$ GeV

and $-\langle\overline{\psi}\psi\rangle_{\text{inv}}^{m_\mu=\epsilon} = (0.218\,\text{GeV})^3$, corresponding to the scale of an up quark. The resulting condensate $\langle\overline{\psi}\psi\rangle_{\text{inv}}^{m_\mu}$ as a function of the current quark mass is illustrated in Fig. 4. Remarkably, the condensate changes sign near the scale $\Lambda_{\text{QCD}}$. For larger masses the absolute value of the condensate increases monotonically. The shown results can be applied to quarks of the SM as well. For example, for a bottom quark we obtain $\langle\overline{\psi}\psi\rangle_{\text{inv}} \approx (4.12\,\text{GeV})^3$. This agrees within a factor two with earlier work [36] which however calculated a slightly different quantity (see discussion before Eq. (14)). For a current mass $m_\mu = 1\,\text{TeV}$ we find $\langle\overline{\psi}\psi\rangle_{\text{inv}} \approx (415\,\text{GeV})^3$. The apparent power law behavior for masses $m_\mu > 1\,\text{GeV}$ allows us to infer the general empirical relation

$$\langle\overline{\psi}\psi\rangle_{\text{inv}} = (c_1\,\text{GeV})^{3-c_2} \times m_\mu^{c_2} \approx (3.7\,\text{GeV})^{1/2}\, m_\mu^{5/2}, \qquad (16)$$

where $c_1$ and $c_2$ are two dimensionless constants and the numerical values in the last step apply to the case of VL triplet quarks.[2] In the following we will generalize our discussion to VL quarks in larger QCD representations.

## 3 Exotic Quarks in Higher Color Representations

Let us now discuss the impact of higher color fermion representations on dynamical chiral symmetry breaking. Therefore, we solve the DSE for chiral quarks up to the **15**-dimensional representation (with Dynkin labels $(2,1)$, see Tab. 1) assuming the effective running coupling of Eq. (3). Numerical results for the chiral condensates found by evaluating Eq. (5) are displayed in Tab. 1. Here we use $\gamma_m = 3C_2(\mathbf{R})/\left[\frac{11}{3}C_2(\mathbf{8}) - \frac{2}{3}n_F\right]$, see e.g. [37]. Evidently, the strength of the condensates increase with the dimension (more precisely, with the increasing quadratic Casimir invariant) of the representations. However, we can not confirm the generation of largely separated scales, as hypothesized by [11–14]. To thoroughly include the effects of fermions in higher dimensional color representations at higher scales (in particular at scales larger than the mass threshold $m_\mu$) it is, of course, necessary to include their effect on the running of $\alpha_s$ itself. At the one loop level, the perturbative running is given by

$$\alpha_s(p^2) = \frac{1}{2\pi\, b\, \ln\left(\frac{p^2}{\Lambda_{\text{QCD}}^2}\right)}, \qquad (17)$$

where $b = \left(\frac{11}{3}C_2(\mathbf{8}) - \frac{2}{3}n_F - \frac{4}{3}T(\mathbf{R})\,n_V\right)/(8\pi^2)$, with $n_V$ being the number of active VL fermions in a representation $\mathbf{R}$ with Dynkin index $T(\mathbf{R})$, while $n_F$ is the number of flavors in the fundamental representation. In the SM with $n_F = 6$ active quarks, $b = 7/(8\pi^2) > 0$ and the strong interaction is asymptotically free. The addition of VL fermions contributes negatively to $b$. In fact, as a result of the increasing Dynkin index, asymptotic freedom is already lost by adding a **10**-plet. Nonetheless, it is possible to include up to two **6**-plets or one **8**-plet fermion while still preserving asymptotic freedom of QCD (see Fig. 5). For the asymptotically free cases of **6** and **8**-plets we have repeated the analysis of the previous section to determine the condensate as a function of the current quark mass. The resulting empirical values for the constants $c_1$ and $c_2$ are collected in Tab. 1. Also for higher dimensional representations the power law behavior $\langle\overline{\psi}\psi\rangle_{\text{inv}} \propto m_\mu^{5/2}$ turns out the be a good approximation.

Altogether, we see that there is not much freedom to add many quarks in higher dimensional representations without running into a strong coupling regime at new high energy

---

[2]We have fit the general relation in Eq. (16) to our result for $\langle\overline{\psi}\psi\rangle_{\text{inv}}$ obtained with the two different methods. We obtain $(c_1, c_2) = (3.71,\ 2.53)$ for method 1, and $(c_1, c_2) = (44.40,\ 2.66)$ for method 2. This shows that $c_2$ is sufficiently close to $5/2$, which is the value we adopt for our approximation in Eq. (16).

Table 1: Summary of our results for the chiral and vector-like condensates for quarks in different representations of QCD. We show Dynkin labels $(p, q)$, quadratic Casimir invariants $C_2(\mathbf{R})$ and Dynkin indices $T(\mathbf{R})$ of the representations, as well as the corresponding mass anomalous dimensions $\gamma_m$. The renormalization group invariant chiral condensates $\langle \overline{\psi}\psi \rangle_{\text{inv}}$ are obtained from Eq. (5). The last two columns describe the VL quark condensates $\langle \overline{\psi}\psi \rangle_{\text{inv}}^{m_\mu}$ as a function of the current quark mass $m_\mu$ in the limit $m_\mu \gg \Lambda_{\text{QCD}}$, in terms of the fit values $c_1$ and $c_2$ of the general empirical relation Eq. (16) for two different extraction methods (see text). The full solution for a VL quark in the triplet representation is shown in Fig. 4. We do not include representations larger than the **8** here, as the strong interaction then turns non asymptotically free (n.a.f.).

| **Rep R** | $(p, q)$ | $C_2(\mathbf{R})$ | $T(\mathbf{R})$ | $\gamma_m$ | $-\langle \overline{\psi}\psi \rangle_{\text{inv}}$ | $(c_1, c_2)$ Method 1 | $(c_1, c_2)$ Method 2 |
|---|---|---|---|---|---|---|---|
| **3** | (1, 0) | 4/3 | 1/2 | 12/21 | $(0.218\,\text{GeV})^3$ | $(3.71, 2.53)$ | $(44.40, 2.66)$ |
| **6** | (2, 0) | 10/3 | 5/2 | 30/21 | $(0.337\,\text{GeV})^3$ | $(62.32, 2.30)$ | $(163.73, 2.41)$ |
| **8** | (1, 1) | 3 | 3 | 27/21 | $(0.363\,\text{GeV})^3$ | $(91.71, 2.34)$ | $(317.52, 2.45)$ |
| **10** | (3, 0) | 6 | 15/2 | 54/21 | $(0.485\,\text{GeV})^3$ | n.a.f. | n.a.f. |
| **15** | (2, 1) | 16/3 | 10 | 48/21 | $(0.521\,\text{GeV})^3$ | n.a.f. | n.a.f. |

scales. We want to avoid this situation in the following and, instead, focus on a minimal setting with only one heavy VL quark in the fundamental representation. This minimal setting will be enough to generate a new infrared scale as the expectation value of the fermion

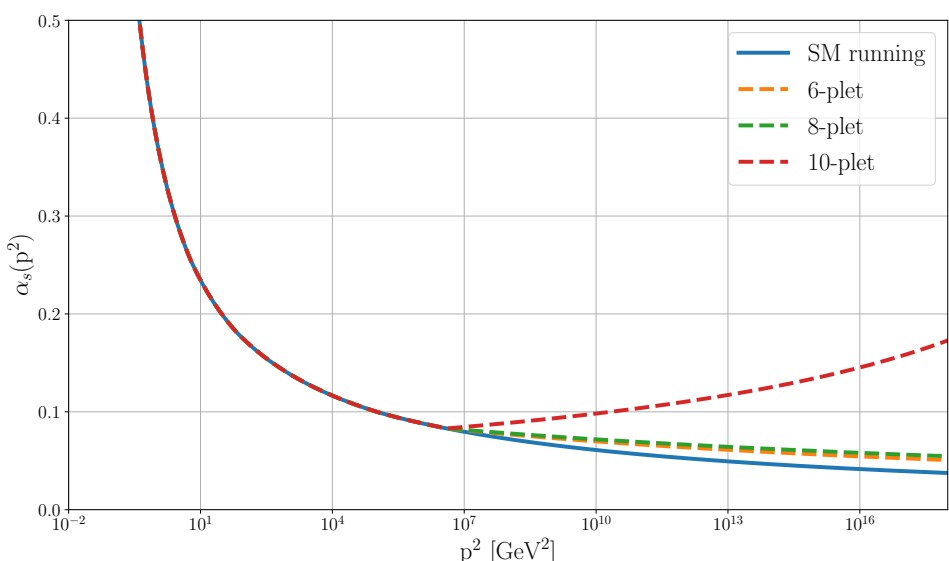

Figure 5: Perturbative one-loop running coupling $\alpha_s(p^2)$ for the SM particle content (blue line) and for the case of SM+a single additional 1 TeV vector-like fermion in a higher dimensional representation of $SU(3)_{\text{C}}$.

two-point function (the condensate) which can be communicated to the Higgs field in order to explain the observed EW scale of the Standard Model. While this requires introducing a new mass scale of the vector-like fermions, it avoids the usual fine-tuning of the hierarchy problem because this new scale is protected by chiral symmetry.

## 4 Inducing the Electro-Weak Scale

In order to transfer the scale of a heavy quark condensate to the EW sector, we introduce a scalar singlet $S$ which couples to both the new VL quark $\psi$ and the SM Higgs field $\phi$. Under $SU(3)_\mathrm{C} \times SU(2)_\mathrm{L} \times U(1)_\mathrm{Y}$ the scalar $S$ transforms as $(\mathbf{1}, \mathbf{1}, 0)$ while the VL quark is assigned to the $(\mathbf{R}, \mathbf{1}, 0)$ representation. There is no principal objection in assigning non-zero hypercharge (hence, non-zero electric charge) to the new quarks but we choose to discuss the simplest incarnation of our mechanism here. The interactions of the VL quark are then described by the Lagrangian

$$\mathcal{L}_\mathrm{VLF} = \overline{\psi}\left(i\slashed{D} - m_\psi - y\,S\right)\psi\,, \tag{18}$$

where $m_\psi$ is the current mass and $y$ denotes the Yukawa coupling to the scalar $S$. We emphasize that with the assigned quantum numbers a direct coupling of the VL quark to the SM color triplet quarks is forbidden. Assuming scale invariance in the scalar sector, the scalar potential is given by

$$V(\phi, S) = \lambda_\phi(\phi^\dagger\phi)^2 + \frac{1}{4}\lambda_S S^4 - \frac{1}{2}\lambda_{\phi S}S^2(\phi^\dagger\phi)\,, \tag{19}$$

with the quartic couplings $\lambda_\phi$ and $\lambda_S$ and the scalar portal coupling $\lambda_{\phi S}$. As shown in the previous sections, even (and in particular) for high current masses $m_\psi$ the VL quark will develop a chiral condensate in the infrared. The condensate eventually breaks scale invariance and acts as a source term which induces a tadpole for $S$. In terms of a tree level effective potential this can be written as

$$V_\mathrm{eff}(\phi, S) = \lambda_\phi(\phi^\dagger\phi)^2 + \frac{1}{4}\lambda_S S^4 - \frac{1}{2}\lambda_{\phi S}S^2(\phi^\dagger\phi) - y\langle\overline{\psi}\psi\rangle_\mathrm{inv}S\,. \tag{20}$$

Hence, the scalar $S$ generically acquires a VEV that is subsequently also transmitted to the Higgs boson via the scalar portal term. This triggers EWSB. It is worth mentioning that the sign of the condensate is not relevant for this mechanism to work as it can always be compensated by the sign of $y$. For the successful development of non-zero expectation values of both scalars, the potential must satisfies the stability conditions $4\lambda_\phi\lambda_S > \lambda_{\phi S}^2$, $\lambda_\phi > 0$ and $\lambda_S > 0$. In unitary gauge the scalar fields are given by

$$\phi(x) = \frac{1}{\sqrt{2}}\begin{pmatrix} 0 \\ v + h(x) \end{pmatrix}, \qquad S(x) = w + s(x)\,, \tag{21}$$

where $v$ and $w$ denote the two VEVs which can be obtained from minimization of Eq. (20) and are given by

$$w^2 = \left(\frac{4\,y\,\lambda_\phi\,\langle\overline{\psi}\psi\rangle_\mathrm{inv}}{4\,\lambda_\phi\,\lambda_S - \lambda_{\phi S}^2}\right)^{\frac{2}{3}}, \qquad \frac{v^2}{w^2} = \frac{\lambda_{\phi S}}{2\lambda_\phi}\,. \tag{22}$$

Using the effective potential, the scalar mass matrix in the $(h, s)$ basis is given by

$$\mathcal{L}_\mathrm{mass} = \frac{1}{2}(h, s)\begin{pmatrix} 2\lambda_\phi v^2 & -\lambda_{\phi S}vw \\ -\lambda_{\phi S}vw & 3\lambda_S w^2 - \frac{1}{2}\lambda_{\phi S}v^2 \end{pmatrix}\begin{pmatrix} h \\ s \end{pmatrix}\,. \tag{23}$$

Diagonalization yields the physical mass eigenstates $(H_1, H_2)$ which are, in the limit $w \gg v$, given by

$$m_{H_1}^2 \approx \left(2\lambda_\phi - \frac{\lambda_{\phi S}^2}{3\lambda_S}\right) v^2,$$
$$m_{H_2}^2 \approx 3\lambda_S w^2,$$
(24)

with a mixing angle

$$\tan(2\theta) \approx \frac{2\lambda_{\phi S} v}{3\lambda_S w}.$$
(25)

The lighter mass eigenstate can be identified as the SM Higgs boson.

By using the empirical relation Eq. (16), we can relate the scale of the VL quark masses of this model to the EW scale as

$$m_\psi \approx \frac{v^{\frac{6}{5}}}{(3.7\,\text{GeV})^{\frac{1}{5}}} \times \left(\frac{2\lambda_\phi}{\lambda_{\phi S}}\right)^{\frac{3}{5}} \left(\frac{4\lambda_\phi\lambda_S - \lambda_{\phi S}^2}{4y\lambda_\phi}\right)^{\frac{2}{5}}.$$
(26)

Furthermore, there is a relation between the new physical scales of the model given by

$$m_\psi \approx \frac{m_{H_2}^{\frac{6}{5}}}{(3.7\,\text{GeV})^{\frac{1}{5}}} \times \left(\frac{1}{3\lambda_S}\right)^{\frac{3}{5}} \left(\frac{4\lambda_\phi\lambda_S - \lambda_{\phi S}^2}{4y\lambda_\phi}\right)^{\frac{2}{5}}.$$
(27)

We illustrate this correlation between the two new masses in Fig. 6, where we use a set of $8 \times 10^4$ random couplings in the reasonable range $y, \lambda_S, \lambda_{\phi S} \in [0.1, 1.5]$ and $\lambda_\phi = 0.13$ fixed to the value inferred for the SM [16]. Varying the scalar and Yukawa couplings in these intervals implies a prediction of the VL quark mass in a range $1 \div 3000\,\text{GeV}$.

To show that our model can successfully reproduce the observed Higgs mass, we give here an exemplary benchmark point. Choosing the color triplet representation and $m_\psi = 1.5\,\text{TeV}$ we find from our full numerical analysis $\langle\overline{\psi}\psi\rangle_{\text{inv}} \approx (590\,\text{GeV})^3$. Using the couplings

$$\lambda_\phi = 0.130, \qquad \lambda_S = 0.695, \qquad \lambda_{\phi S} = 0.100, \qquad y = 0.210,$$
(28)

the exact diagonalization of the mass matrix in Eq. (23) results in scalar masses

$$m_{H_1} = 125.1\,\text{GeV}, \qquad \text{and} \qquad m_{H_2} = 574.7\,\text{GeV},$$
(29)

with a mixing angle $\tan(2\theta) = 6.3 \times 10^{-2}$. Even though the new scalar $H_2$ has sub-TeV mass at this benchmark point, it is not excluded by constraints from LHC data [38].

Experimental limits on VL quarks crucially depend on the their representation assignment. Direct searches at the LHC usually consider either EW doublets or EW singlets with the same electric charges as up- or down-type quarks, decaying into SM quarks under the emission of $W, Z$ or Higgs bosons. They constrain VL quark masses to be heavier than approximately 1.5 TeV [39,40] but apply to our model only if we assign the VL quark a non-zero hypercharge. For our benchmark scenario of zero hypercharge, the VL quarks would be stable, but could still be pair-produced to form exotic hadrons which results in similar limits on their mass [41,42]. Hence, $m_\psi \gtrsim 1.5\,\text{TeV}$ may be regarded as a conservative lower limit on the VL current quark mass.

In the case of zero hypercharge, neutral hadrons made out of the heavy VL quarks would, in fact, be good, colored Dark Matter (DM) candidates [43]. The treatment in [43] mainly focussed on quarks in the representation $(\mathbf{8}, \mathbf{1}, 0)$, which would also work for our present mechanism, and assumed the absence of condensates. Our present work would then also motivate a more detailed look at the DM phenomenology of the $(\mathbf{3}, \mathbf{1}, 0)$ VL quarks along the lines of [43], and under the inclusion of condensates.

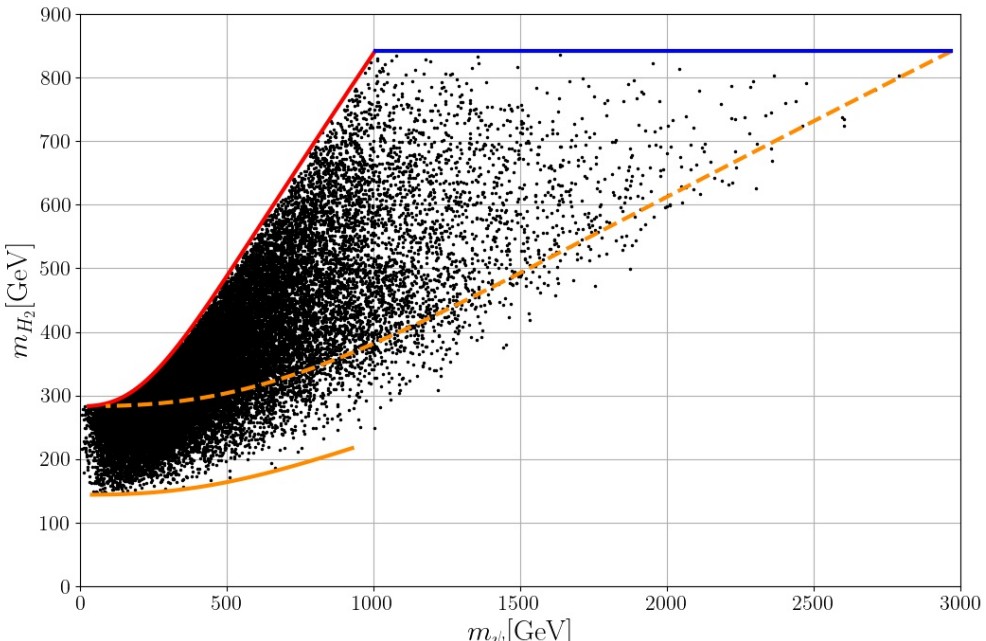

Figure 6:    Correlation between the new physics scales $m_\psi$ and $m_{H_2}$ following from Eqs. (26) and (27) for a set of $8 \times 10^4$ random combinations of couplings $y, \lambda_S, \lambda_{\phi S} \in [0.1, 1.5]$ and fixed $\lambda_\phi = 0.13$. The blue line indicates the upper limit on $m_{H_2}$ which is achieved for $\lambda_S = 1.5$ and $\lambda_{\phi S} = 0.1$, while the red envelope shows the behavior for $y = 1.5$, $\lambda_S = 1.5$ and $\lambda_{\phi S}$ varying between 0.1 to 1.5. The boundary to lower values of $m_{H_2}$ is obtained as a superposition of curves with varying $\lambda_S$ and $\lambda_{\phi S}$ with fixed $y = 0.1$. We show two of this curves for $\lambda_S = 0.1$ (solid orange) and $\lambda_S = 1.5$ (dashed orange).

## 5    Discussion and Conclusion

In order to obtain the dynamical wave function renormalization and mass function of a heavy VL quark, we have numerically solved the Dyson-Schwinger equation in rainbow approximation, using a phenomenologically motivated effective running QCD coupling. We have used our solutions to extract the condensate (the expectation value of the fermion two-point function) as a function of the VL quark QCD representation and current quark mass. Using two different methods, which agree in their result, we have extracted the numerical value of the renormalization group invariant condensate.

Our computation allows us to reproduce the well-known values of the chiral condensate for zero current quark mass, and for the light quark masses. Furthermore, we find that the condensate has a zero-crossing around the QCD scale, after which it monotonically grows as a function of the current quark mass. Hence, for heavy VL quarks the dynamically generated expectation value of their two point function corresponds to large infrared scales. If the VL quark couples to a scalar field, this will effectively generate a tadpole in the scalar potential which can induce a VEV of the scalar field despite its otherwise scale invariant potential. If the scalar couples to the SM Higgs portal, this VEV can serve to dynamically generate the EW scale and EWSB.

We have presented the simplest realization of this mechanism which requires a new SM

singlet scalar field as well as a VL quark in a low-dimensional representation of QCD. We have given a set of benchmark parameters that reproduces the observed Higgs mass while predicting a new neutral scalar field close to the EW scale, as well as a VL quark around the TeV scale. In our simplest model, the relation

$$m_{H_2} \approx \left( \frac{6 \lambda_\phi \lambda_S}{\lambda_{\phi S}} \right)^{1/2} \times v \,, \tag{30}$$

shows that we cannot decouple the new neutral scalar without fine-tuning scalar couplings, which might be considered unnatural. By contrast, the fact that the VEV of the new scalar depends on the VL mass scale proportional to a Yukawa coupling,

$$w \propto y^{1/3} \langle \overline{\psi}\psi \rangle_{\text{inv}}^{1/3} \propto y^{1/3} m_\mu^{5/6} (\text{GeV})^{1/6} \,, \tag{31}$$

shows that we can, in principle, decouple the VL quarks without spoiling the mechanism if we allow for a small Yukawa coupling $y$. Such a smallness can be considered natural as it is protected by the chiral symmetry of the new quarks.

Interestingly, if the VL quarks have zero hypercharge they are stable and form baryons which have previously been discussed as very well suited, colored Dark Matter candidates [43] and it will be interesting to see how the presence of condensates modifies this discussion. Hence, our model could economically solve two of the most pressing puzzles in our understanding of Nature.

Finally, we note that while in our simplest setup classical conformal symmetry is explicitly broken only by the rigid VL mass scale, our discussion can straightforwardly be extended to other variations. On the one hand, including a VL condensate opens new parameter space for EWSB because it can trigger spontaneous symmetry breaking even in the presence of scalar mass terms with the usually "wrong" sign. On the other hand, our findings also reopen the door to new, entirely conformal solutions to the EW hierarchy problem. This is straightforward to realize by modifying our model in such a way that the current quark mass itself is generated from yet another scale invariant scalar sector by spontaneous symmetry breaking á la Coleman-Weinberg.

# Acknowledgments

We would like to thank Fei Gao for useful discussions.

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
