# Peer review of "Generating the Electro-Weak Scale by Vector-like Quark Condensation"

_SciPost Physics, doi:SciPost Phys. 14, 076 (2023)_

## Round 1 · Referee Report · Anonymous · 2022-9-25

Strengths
1. Manuscript contains calculations of quark condensates that would be generally applicable to other BSM models.
2. Writing style is clear.
3. The BSM model introduced in the manuscript is simple with many signatures.
Weaknesses
1. The role played by different ingredients within the BSM model presented in the manuscript are not clearly disentangled.
Report
The manuscript "Generating the Electro-Weak Scale by Vector-like Quark Condensation" presents a new model of the electroweak sector, in which the standard model is extended with a vector like quark and scalar singlet. In this new model, the scalar sector is scale invariant at the classical level, and the electroweak scale is generated dynamically, as a result of the vector like quark $\psi$ acquiring a nonvanishing condensate $\langle\bar{\psi}\psi\rangle$. The condensate is determined by solving Dyson-Schwinger equations approximately, which incorporates effects from QCD. The results of this calculation are then used to identify a viable parameter space for the model, consistent with experimental constraints.
The model presented in the manuscript is simple, and yet has a variety of experimentally testable signatures. Furthermore, the results for the $\langle\bar{\psi}\psi\rangle$ condensate presented in Fig 4, and extended to vector like quarks in higher representations of QCD color in section 3 might have general applicability to many other beyond-standard-model theories. The paper therefore meets expectation 3 of the acceptance criteria and I am therefore inclined to recommend publication. However, there are a couple of conceptual issues which would need to be addressed in the manuscript first.
Firstly - is it essential that the new vector like fermion carries QCD color for the model to break electroweak symmetry and be phenomenologically viable? Any uncolored massive fermion having a yukawa coupling to the scalar sector should induce (perturbatively) a Coleman Weinberg potential for the scalars that could trigger dynamical electroweak symmetry breaking. Also, from the presentation, it would seem that an uncolored massive fermion would acquire a large condensate perturbatively as shown in Eq 8. Even if the quadratically divergent contribution to the condensate is subtracted away, a log divergent part would remain, which grows as fermion mass cubed. This is a very similar scaling behavior for the condensate with fermion mass to that suggested in Eq 16. The role that QCD is playing in the mechanism should be clarified. If the model with an uncolored vector like fermion would also be viable, this needs to be clearly stated in the introduction and section 4.
Secondly, to what extent could the results for the condensate in figure 4 be applied to the bottom and top quarks of the standard model? This should at least be commented on, to enable comparison with results for these condensates obtained using different methods.
Author: Sophie Klett on 2022-11-07 [id 2989]
(in reply to Report 1 on 2022-09-25)Dear referee,
For ease of reading, we wrote our response as a pdf file which is attached to this comment.
Sincerely,
The authors
Attachment:
referee_answer.pdf

---

## Round 2 · Referee Report · Anonymous (Referee 1) · 2022-12-16

Report

All of my questions have been resolved in the latest version of the authors' manuscript, which is well written. The essential role of QCD within the mechanism that underlies their model is now clear. Overall, the manuscript provides a compelling picture for a possible dynamical origin of the electroweak symmetry breaking sector.

---

## Round 2 · Author Response

Dear editor and dear referee,

We are grateful for the close reading and insightful comments. Below we list the changes we have made in the revised manuscript.
A detailed response to the referee report is posted under the "Reply to the above Report" section below the report.

Sincerely,
The authors

---

## Round 2 · List of Changes

1. We have added a footnote on page 3 to emphasize the importance of a strongly coupled theory (QCD) in the mechanism.

  2. To further clarify the cancellation of divergences in the calculation of a massive fermion condensate we have added a sentence before Eq. (12) on page 7.

  3. We have added a sentence to page 9 of our updated manuscript to comment on the applicability of our results to quarks of the standard model.

---

## Editorial Decision

published